# Sex-stratified pharmacovigilance of gastrointestinal events associated with first-line smoking-cessation medicines: Insights from the FAERS database

Haoxiong Sun[1,2☯*], Junchi Chen[3☯], Xiaoxuan Wu[4], Ziyan Wang[5]

1 Department of Behavioural Science and Health, University College London, London, United Kingdom, 2 School of Clinical Sciences at Monash Health, Monash University, Melbourne, Australia, 3 Centre for Medicine Use and Safety, Faculty of Pharmacy and Pharmaceutical Sciences, Monash University, Melbourne, Australia, 4 Department of pharmacy, Uppsala University, Uppsala, Sweden, 5 Department of Infectious Disease, Faculty of Medicine, Imperial College London, London, United Kingdom

☯ These authors contributed equally and shared first authorship.
* haoxiong.sun.24@ucl.ac.uk

## Abstract

### Background

Tobacco smoking is a major global health threat. Pharmacological aids, including nicotine-replacement therapy (NRT), varenicline, and bupropion, improve quit rates but are associated with gastrointestinal (GI) adverse events (AEs) that can compromise adherence. The real-world reporting profiles of these GI AEs, particularly the differences between sexes, have not been comprehensively characterized.

### Methods

We analyzed the FDA Adverse Event Reporting System (FAERS) database from 2004 Q1 to 2024 Q2. After deduplication, reports designating NRT, varenicline, or bupropion as the primary suspect drug were extracted. Disproportionality analyses, including the Proportional Reporting Ratio (PRR) and Reporting Odds Ratio (ROR), were conducted to quantify drug-event associations. The Breslow–Day test was used to assess the homogeneity of RORs between male and female strata.

### Results

Varenicline was associated with the highest proportion of GI reports (36.0% of its total reports). The disproportionality signal was significantly stronger in women than in men (ROR 6.41 vs. 5.10 for nausea, $p < 0.001$). NRT was linked to 24.3% of GI reports, with hiccups (PRR = 60.1) being the most prominent signal. In contrast to varenicline, several key GI AE signals for NRT were significantly stronger in men (e.g., nausea, ROR 3.09 in men vs. 2.45 in women, $p < 0.001$). Bupropion had the

**Data availability statement:** All raw individual case safety reports analysed in this study are freely available from the U.S. Food and Drug Administration Adverse Event Reporting System (FAERS) public dashboard (https://fis.fda.gov/extensions/FPD-QDE-FAERS/FPD-QDE-FAERS.html).

**Funding:** The author(s) received no specific funding for this work.

**Competing interests:** The authors have declared that no competing interests exist.

lowest proportion of GI reports (2.1%) but still generated significant disproportionality signals (overall ROR 4.50), particularly for anorexia (PRR = 4.80) and dry mouth (PRR = 4.42), with most signals being stronger in women.

## Conclusion

NRT, varenicline, and bupropion exhibit distinct and statistically significant sex-specific GI AE reporting profiles in a real-world setting. These hypothesis-generating findings underscore the importance of considering sex as a variable in pharmacovigilance studies and may inform future research aimed at personalizing smoking cessation therapy.

## Introduction

Tobacco smoking, a leading cause of preventable disease and death globally, contributes to about 8 million deaths annually through its effects on cardiovascular, pulmonary and mental health [1–3]. Smoking cessation has been shown to be associated with numerous health benefits, and clinical guidelines recommend pharmacological aids, primarily nicotine-replacement therapy (NRT), varenicline and bupropion as first-line interventions to be used in conjunction with behavioral support [4,5]. These pharmacological agents have demonstrated efficacy in enhancing smoking cessation rates and currently prescribed to millions of tobacco users annually.

However, those pharmacotherapies are associated with adverse events (AEs) which may reduce patients' adherence and ultimately impact quit rates. Gastrointestinal (GI) symptoms are among the most frequently reported adverse effects, even though they generally demonstrate less severe clinical consequences compared to other side effects. Some studies reviewed and examined the physiological basis for these digestive tract AEs. [6,7] For example, nicotine administered via NRT activates nicotinic acetylcholine receptors (nAChRs), which may lead to gastric response like dysregulation of acid secretion [8]. Varenicline, while developed as a partial $\alpha 4 \beta 2$ nAChR agonist, demonstrates potent agonistic activity at 5-HT3 receptors, potentially lead to GI tract side effects [9]. Bupropion is a norepinephrine–dopamine re-uptake inhibitor with mild anticholinergic and negative allosteric effects on certain nAChR subtypes, and it can slow intestinal transit and reducing secretions [10].

Both randomized controlled trials (RCTs) and regulatory product labels acknowledge the occurrence of GI adverse events, for example, RCTs have consistently reported nausea as the most common adverse event with varenicline, while bupropion is more often associated with dry mouth. Observational studies from large health databases have also documented gastrointestinal complaints, though with varying incidence estimates [8,11–13]. However, these studies are limited in their ability to detect rare or unexpected adverse events, and clinical trials are typically conducted under controlled conditions that may not reflect real-world clinical practice. These limitations indicate the value of post-marketing surveillance resources. The Food and Drug Administration Adverse Event Reporting System (FAERS) provides

a pharmacovigilance resource for large-scale post-marketing surveillance, while current studies mainly focus on general side effects rather than focusing on specific GI tract disease [7,14,15].

An additional research gap is the limited investigation of sex-related differences in adverse event reporting. Women report nearly twice as many adverse drug events as men and appear especially susceptible to GI AEs (Manteuffel et al., 2014). They also experience additional tobacco-related harms and, during cessation, face sex-specific challenges such as higher rates of depression and a stronger physiological response to nicotine reduction (Zucker & Prendergast, 2020, Smith et al., 2017). These differences indicate the importance of sex-stratified research on GI adverse events, to inform tailored counselling, minimize treatment discontinuation and ultimately improve quit outcomes.

The present study addresses these gaps by using the FAERS database to examine reports submitted from 2004 to the second quarter of 2024. The aim was to identify evidence of a disproportionate association among varenicline, bupropion, or NRT and gastrointestinal adverse events. In addition, the study characterizes GI safety signals across these pharmacotherapies, describes the demographic profiles of patients reporting the most frequent events, and presents sex-stratified analyses. The findings highlight sex-specific reporting patterns, enhance understanding of post-marketing safety signals, and generate hypotheses for future research.

## Method

### Data source

This pharmacovigilance analysis utilized the FDA Adverse Event Reporting System (FAERS) as the data source for adverse event reports spanning from the first quarter of 2004 to the second quarter of 2024. The data was accessed on March 20$^{Th}$ 2025, and authors did not have access to any personally identifiable information during or after data collection. We extracted all individual case safety reports (ICSRs) associated with smoking cessation therapies using OpenVigil 2.1 – an established platform for structured FAERS data mining. All adverse events terms in the dataset were coded according to the Medical Dictionary for Regulatory Activities (MedDRA) version 27.0 terminology, to ensure terminological consistency with System Organ Class (SOC) and Preferred Term (PT). For each ICSRs, we captured relevant fields including patient demographic data, including sex, age, and country of origin, the reported adverse events (MedDRA PTs), and the clinical outcomes.

### Ethical statements

This study utilized de-identified, publicly available data from the FAERS, which does not involve direct interaction with human participants. As such, ethical approval and informed consent were not required.

### Data filtering

Reports were identified using drug name searches in both the drugname and prod_ai fields of FAERS. Generic names, brand names, and formulation-specific terms were combined with OR operators to maximize capture (**Table 1**). The search strategy followed established pharmacovigilance methods [7].

All reports in which varenicline, bupropion, or NRT products were designated as a suspect or interacting drug were included. GI AEs were defined primarily as Preferred Terms under the MedDRA System Organ Class (SOC) 'Gastrointestinal disorders.' In addition, a small number of oral and digestive tract–related terms frequently associated with smoking-cessation medications (e.g., dry mouth, hiccups, dental caries) were also included, based on prior pharmacovigilance literature and clinical relevance (Motooka et al., 2018). The full list of PTs is provided in **S1 Table**.

To ensure data integrity and eliminate redundancy, rigorous deduplication process following FDA recommendations were applied. Multiple records referring to the same case were identified by a common case number or Individual Safety Report (ISR) identifier. After identification, they were consolidated by retaining only the most recent version (based on report date or version number). Reports that were exact duplicates or shared the same combination of case identifiers,

**Table 1. Search terms used to identify smoking-cessation medications in FAERS. Queries were applied with OR operators across the drugname and prod_ai fields. Reports for Wellbutrin were excluded to minimize confounding by non-cessation indications (e.g., depression).**

| Medication | Search terms (drugname/prod_ai fields) |
| --- | --- |
| NRT | nicotine; Nicoderm CQ; Nicorette; Habitrol; Nicotrol; Thrive; Nicorelief; nicotine patch; gum; lozenge; inhaler; nasal spray; polacrilex |
| Varenicline | varenicline; Chantix; Champix |
| Bupropion | bupropion; Zyban; bupropion SR |

report date, drug name, indication, sex, country, and age were excluded to ensure each case was counted once. In addition, we removed cases in which the GI AE was explicitly attributed by the reporter to drug–drug interactions or concomitant medications, rather than to the primary suspect drug. After applying all filtering and exclusion criteria, a final dataset of unique FAERS reports of GI AEs linked to the target medications was generated for analysis.

## Data mining

A disproportionality analysis was conducted to identify potential safety signals of GI adverse events associated with the three smoking cessation therapies. In particular, two frequentist signal detection algorithms were utilized: the Proportional Reporting Ratio (PRR) and the Reporting Odds Ratio (ROR) [16]. The PRR was calculated as the ratio of the proportion of GI-related adverse events among all reports involving the drug of interest to the proportion among reports for all other drugs. The ROR was defined as the odds of a GI adverse event being reported for the drug of interest relative to the odds of the same event being reported for all other drugs. A signal was considered present if a given drug–event (DE) pair had at least three reports and met the threshold of PRR ≥ 2 with an accompanying chi-square ($\chi^2$) value ≥ 4 [17]. Similarly, for the ROR, a signal was seen as significant if the lower bound of its 95% confidence interval was greater than 1 while the DE was at least three. Drug–GI event combinations meeting these disproportionality criteria were identified as signals of a potential association between the medication and the gastrointestinal adverse event.

## Sex-stratified disproportionality and Breslow–Day test

To assess sex differences, all DE pairs were stratified by male and female. Sex-specific RORs were computed from 2 × 2 tables with cells (a,b,c,d) where a = drug with event, b = drug without event, c = other drugs with event, d = other drugs without event. A Haldane–Anscombe 0.5 continuity correction was applied when a cell was zero.

Homogeneity of odds ratios was tested between sexes using the Breslow–Day $\chi^2$ test. To reduce spurious findings from sparse strata, we pre-specified a stability filter of DE_male + DE_female > 20; statistical significance was defined as two-sided $p < 0.05$ (no multiplicity adjustment). Only events meeting both stability and significance criteria are shown in the main forest plot and table.

## Statistical analysis

All data aggregation, statistical analyses, and visualization were conducted using Microsoft Excel 2021 (Microsoft Corp., Redmond, WA, USA) and RStudio Version 4.5.0 (R Foundation for Statistical Computing, Vienna, Austria). Descriptive statistics were used to summarize report characteristics and the proportion of GI adverse events for each drug. These results were visualized using heat maps and forest plots, with significant signals highlighted.

## Result

Following selection processes shown in Fig 1, reports characteristics for each medication were summarized (Table 2). A total of 54,303 primary suspect reports were analyzed across the three smoking-cessation therapies. A majority of reports

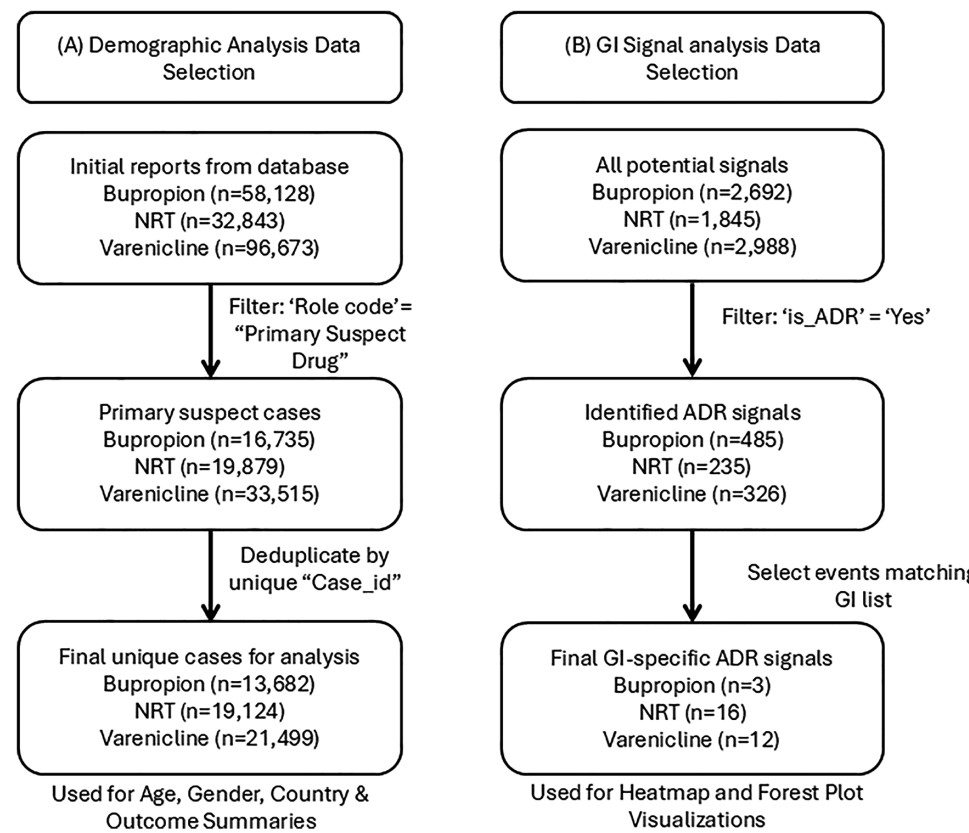

**Fig 1. Flowchart of FAERS case selection for demographic (A) and gastrointestinal (B) analyses.**

involved female patients (61.6%; n = 33,443), while 35.9% involved male patients (n = 19,488); the remainder listed gender as "Other" or "Unknown" (2.5%; n = 1,372).

The age distribution of reporters varied by medication. Reports for varenicline were most concentrated among adults aged 40–55 years (46%; n = 9,883). In contrast, bupropion reports skewed towards a younger demographic, with the highest proportion in the 25–40 year age group (35%; n = 4,798) and a notable percentage under 25 years (11%; n = 1,459). Nicotine replacement therapy (NRT) reports were more evenly distributed across age bands: 25–40 years (25%; n = 4,805), 40–55 years (38%; n = 7,314), and over 55 years (31%; n = 6,017).

Geographically, the vast majority of reports originated from the United States (91.3%; n = 49,570). The United Kingdom (1.2%; n = 639), Canada (1.1%; n = 595), and Japan (0.4%; n = 214) each contributed a small fraction of the total reports.

Reported clinical outcomes demonstrated significant differences between the therapies. Bupropion was associated with the highest proportion of serious outcomes, including death (19%; n = 2,539), life-threatening events (5%; n = 720), and hospitalizations (14%; n = 1,915). Varenicline presented with intermediate levels of serious outcomes. Conversely, NRT was infrequently associated with death (<1%; n = 105), and a substantial majority of its reports (86%; n = 16,521) were categorized with an "unknown outcome."

Overall, each smoking-cessation therapy was associated with a distinct GI AE profile, with detailed signal strengths for individual AEs available in **S2 Table**. Varenicline had the highest proportion of GI-related reports, as shown in **Table 3**. Bupropion's strongest overall signal was for anorexia (PRR = 4.80), while NRT was most prominently associated with hiccups (PRR = 60.1), and varenicline was led by flatulence (PRR = 9.98). A summary of these signals is visualized in the heat maps in **Fig 2**.

**Table 2. Demographic, geographic and outcome characteristics of adverse-event reports for three therapies.** Values are *n* (% of all reports for that drug). "Other countries" aggregates all nations that individually contributed <1% of reports. "Unknown" indicates that the corresponding data field was blank or could not be coded in the source files.

| | NRT (n = 19,124) | Varenicline (n = 21,498) | Bupropion (n = 13,681) |
|---|---|---|---|
| **Gender** | | | |
| Female | 11935(62.41) | 12690(59.03) | 8818(64.45) |
| Male | 6673(34.89) | 8262(38.43) | 4553(33.28) |
| Other | 5(0.03) | 14(0.07) | 16(0.12) |
| Unknown | 511(2.67) | 532(2.47) | 294(2.15) |
| **Age group** | | | |
| <25 | 988(5.17) | 540(2.51) | 1459(10.66) |
| 25-40 | 4805(25.13) | 5000(23.26) | 4798(35.07) |
| 40-55 | 7314(38.25) | 9883(45.97) | 5265(38.48) |
| >55 | 6017(31.46) | 6075(28.26) | 2159(15.78) |
| **Reporting country** | | | |
| United States | 16745(87.56) | 20055(93.29) | 12770(93.34) |
| United Kingdom | 233(1.22) | 279(1.30) | 127(0.93) |
| Canada | 201(1.05) | 241(1.12) | 153(1.12) |
| Japan | 73(0.38) | 86(0.40) | 55(0.40) |
| Other countries | 1639(8.57) | 651(3.03) | 430(3.14) |
| Unknown | 233(1.22) | 186(0.87) | 146(1.07) |
| **Outcome** | | | |
| Death | 105(0.55) | 520(2.42) | 2539(18.56) |
| Life-threatening event | 55(0.29) | 619(2.88) | 720(5.26) |
| Hospitalization | 423(2.21) | 2032(9.45) | 1915(14) |
| Disability | 33(0.17) | 387(1.8) | 275(2.01) |
| Required intervention | 27(0.14) | 144(0.67) | 153(1.12) |
| Other adverse events | 1960(10.25) | 5830(27.12) | 3307(24.17) |
| Unknown | 16521(86.39) | 11966(55.66) | 4772(34.88) |

**Table 3. Disproportionality metrics for gastro-intestinal adverse-event (AE) reports associated with three therapies.** Gastrointestinal AEs as PSn (%) indicates percentage of all primary-suspect reports for that drug that were coded as GI events. All signals met conventional disproportionality criteria ($\chi^2 \geq 4$, minimum three cases, and 95% CI lower bound >1).

| Drug name | Gender subgroup analysis | Gastrointestinal AEs as PSn (%) | ROR (95%CI) | PRR (95%CI) | $\chi^2$ |
|---|---|---|---|---|---|
| NRT | | 24.3 | 4.04 (3.9, 4.18) | 3.41 (3.31, 3.5) | 7091.76 |
| | male | 23.0 | 4.14(3.95, 4.34) | 3.66 (3.52,3.81) | 4204.83 |
| | female | 25.1 | 3.32(3.2,3.43) | 2.93 (2.85, 3.02) | 5311.15 |
| Varenicline | | 36.01 | 4.77 (4.65, 4.88) | 3.41 (3.36, 3.46) | 19550.94 |
| | male | 23.5 | 4.2 (4.09,4.35) | 3.46 (3.38, 3.54) | 9590.03 |
| | female | 40.63 | 5.98 (5.86, 6.11) | 3.96 (3.91, 4.01) | 36692.94 |
| Bupropion | | 2.11 | 4.5 (4.04, 5.03) | 4.43 (3.98, 4.93) | 869.14 |
| | male | 1.6 | 4.79 (4.07, 5.63) | 4.72 (4.03, 5.54) | 434.8 |
| | female | 4.22 | 3.03 (2.83, 3.25) | 2.95, (2.76, 3.15) | 1133.43 |

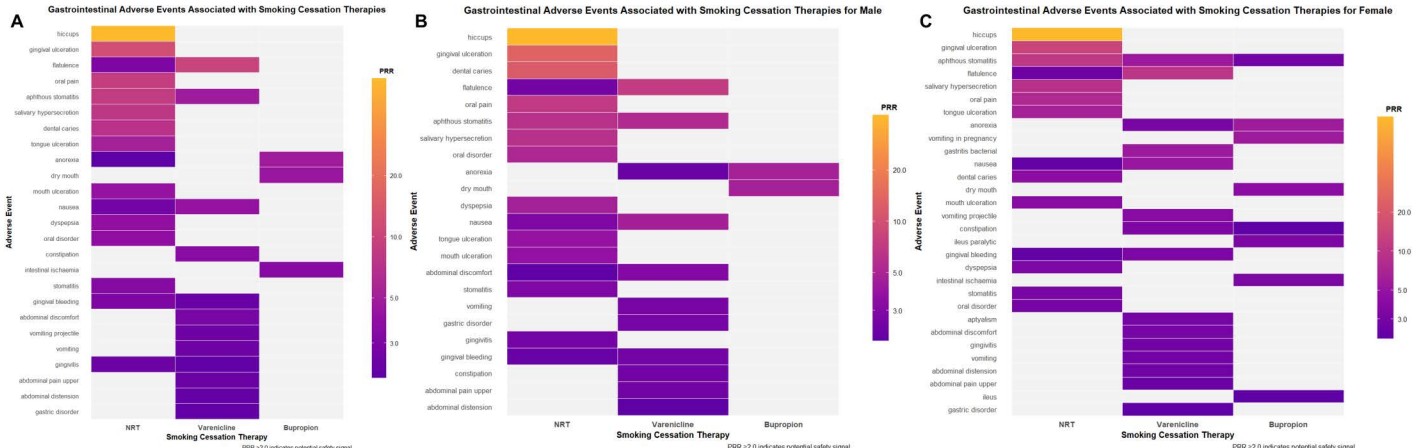

**Fig 2. Heat maps of PRR for GI AEs associated with NRT, Varenicline, and Bupropion.** Panels show (A) overall, (B) male, and (C) female. GI AEs were selected by meeting the dual filter of DE ≥ 3 and PRR ≥ 2. Color grading is log10-scaled; blank cells indicate no qualifying reports.

Formal heterogeneity testing of sex-stratified reports revealed significant and distinct patterns of sex-specific disproportionality for all three drugs, with detailed results presented in **Table 4** and visualized in **Fig 3**. For bupropion and varenicline, the signals showing significant heterogeneity were predominantly stronger in women. Notably, as detailed in **Table 4**, women exhibited significantly higher reporting odds for varenicline-associated nausea (Female ROR 6.41 vs. Male ROR 5.10, p < 0.001) and bupropion-associated constipation (Female ROR 2.21 vs. Male ROR 1.68, p = 0.008).

In contrast, NRT exhibited a more complex profile, with a majority of its heterogeneous signals being significantly stronger in men. This pattern was evident for key GI events such as nausea (Male ROR 3.09 vs. Female ROR 2.45, p < 0.001) and dyspepsia (Male ROR 4.74 vs. Female ROR 3.30, p < 0.001), as shown in **Table 4**. Key exceptions for NRT included hiccups and salivary hypersecretion, for which the signals were significantly stronger in women. These statistically significant sex-specific findings are visualized in **Fig 3**.

## Discussion

Our analysis of the FAERS database revealed distinct reporting profiles for GI AEs among the three main smoking cessation pharmacotherapies. Varenicline was associated with the highest proportion of GI reports (36.0%), followed by NRT (24.3%) and bupropion (2.1%). Although bupropion had the lowest reporting proportion, it still generated strong disproportionality signals (Overall ROR 4.50), underscoring that a low reporting frequency does not preclude a significant safety signal. This study confirms that all three medications are associated with significant GI AE signals relative to the database baseline.

For NRT, the strongest reporting signals were predominantly related to the oral route of administration, with exceptionally high PRRs for contact stomatitis (110.1) and hiccups (60.1). This pattern is biologically plausible, as oral NRT formulations deliver nicotine directly to the mucosal tissues of the oral cavity and upper GI tract. Systemic nicotine exposure may also account for other observed signals, such as nausea, by potentially altering gastric acid secretion and gut motility. Notably, our formal heterogeneity testing revealed a complex sex-specific pattern for NRT; while the signal for hiccups was significantly stronger in women, men exhibited significantly stronger reporting signals for several key GI events, including nausea and dyspepsia (p < 0.001). Different NRT administrate nicotine via different methods, the oral formulation allows nicotine reaches oral and GI tract directly so lead to higher risks of irritating mucosal tissues, which may illustrate the reason of those stronger AEs [18,19]. The systemic nicotine will also increase the acid secretion and alter gut motility [20,21],

**Table 4. Sex-stratified disproportionality analysis of gastrointestinal AEs by Breslow–Day test.**

| Drug | AE | ROR (Male) | ROR (Female) | Breslow–Day p | Direction |
|---|---|---|---|---|---|
| Bupropion | nausea | 1.35 | 1.97 | <0.001 | F > M |
| | vomiting | 0.94 | 1.30 | <0.001 | F > M |
| | abdominal distension | 0.44 | 1.03 | 0.003 | F > M |
| | constipation | 1.68 | 2.21 | 0.008 | F > M |
| | decreased appetite | 0.53 | 0.81 | 0.010 | F > M |
| | flatulence | 0.89 | 1.63 | 0.022 | F > M |
| | haematochezia | 0.20 | 0.56 | 0.037 | F > M |
| NRT | hiccups | 42.70 | 109.81 | <0.001 | F > M |
| | nausea | 3.09 | 2.45 | <0.001 | M > F |
| | abdominal pain upper | 1.86 | 1.05 | <0.001 | M > F |
| | dental caries | 12.06 | 4.32 | <0.001 | M > F |
| | abdominal discomfort | 2.05 | 1.22 | <0.001 | M > F |
| | dyspepsia | 4.74 | 3.30 | <0.001 | M > F |
| | dry mouth | 1.91 | 1.29 | 0.008 | M > F |
| | salivary hypersecretion | 6.16 | 9.14 | 0.034 | F > M |
| Varenicline | nausea | 5.10 | 6.41 | <0.001 | F > M |
| | constipation | 2.51 | 3.41 | <0.001 | F > M |
| | flatulence | 7.45 | 10.16 | <0.001 | F > M |
| | decreased appetite | 0.93 | 1.34 | <0.001 | F > M |
| | abdominal pain lower | 0.81 | 0.22 | <0.001 | M > F |
| | abdominal distension | 2.01 | 2.70 | 0.002 | F > M |
| | gingivitis | 0.53 | 2.79 | 0.017 | F > M |
| | gastric ulcer | 0.37 | 0.89 | 0.040 | F > M |

Sex-specific reporting odds ratios (RORs) are shown for male and female strata. Only events meeting the pre-specified stability filter (DE_male+DE_female > 20) and Breslow–Day p < 0.05 are displayed. Direction indicates which sex had the higher ROR (F > M or M > F).

which explain some of AEs. Overall, NRT exhibits a moderate GI adverse profile with primarily minor symptoms, explaining its lower ROR/PRR signal compared to other cessation medications.

The mechanism of varenicline, a partial agonist at α4β2 nicotinic acetylcholine receptors, may contribute to the high reporting frequency of GI symptoms [9]. T Consistent with this, the main GI AE signals detected for varenicline in our study included flatulence, nausea, and constipation. The strong disproportional reporting for nausea in FAERS aligns with clinical trials, which report incidence rates of 25–40% [22]. This effect results from varenicline's stimulation of central and peripheral nicotinic receptors, including those in the chemoreceptor trigger zone and vague nerve, leading to nausea, vomiting, and dyspepsia [23]. Clinical trials demonstrated an **increase in the odds** of nausea versus placebo, aligning with results from this study [24]. The disproportionate reporting of nausea and vomiting in FAERS confirms varenicline's tendency to cause GI AEs through its "nicotine-like" stimulus mechanism. Furthermore, our study provides real-world evidence of a significant sex difference for this key AE; the reporting signal for nausea was significantly stronger in women than in men (ROR 6.41 vs. 5.10, p < 0.001). A similar female-predominant pattern was also observed for other AEs like constipation and flatulence.

Bupropion functions as not only smoking cessation medication but also an atypical antidepressant by inhibiting norepinephrine-dopamine reuptake [10]. Those dual mechanisms provide potential mechanisms under AEs. Consistent with previous research suggesting a link between bupropion and nausea or vomiting, our analysis not only detected these signals but also revealed a significant sex-specific pattern, with reporting signals for both AEs being significantly stronger

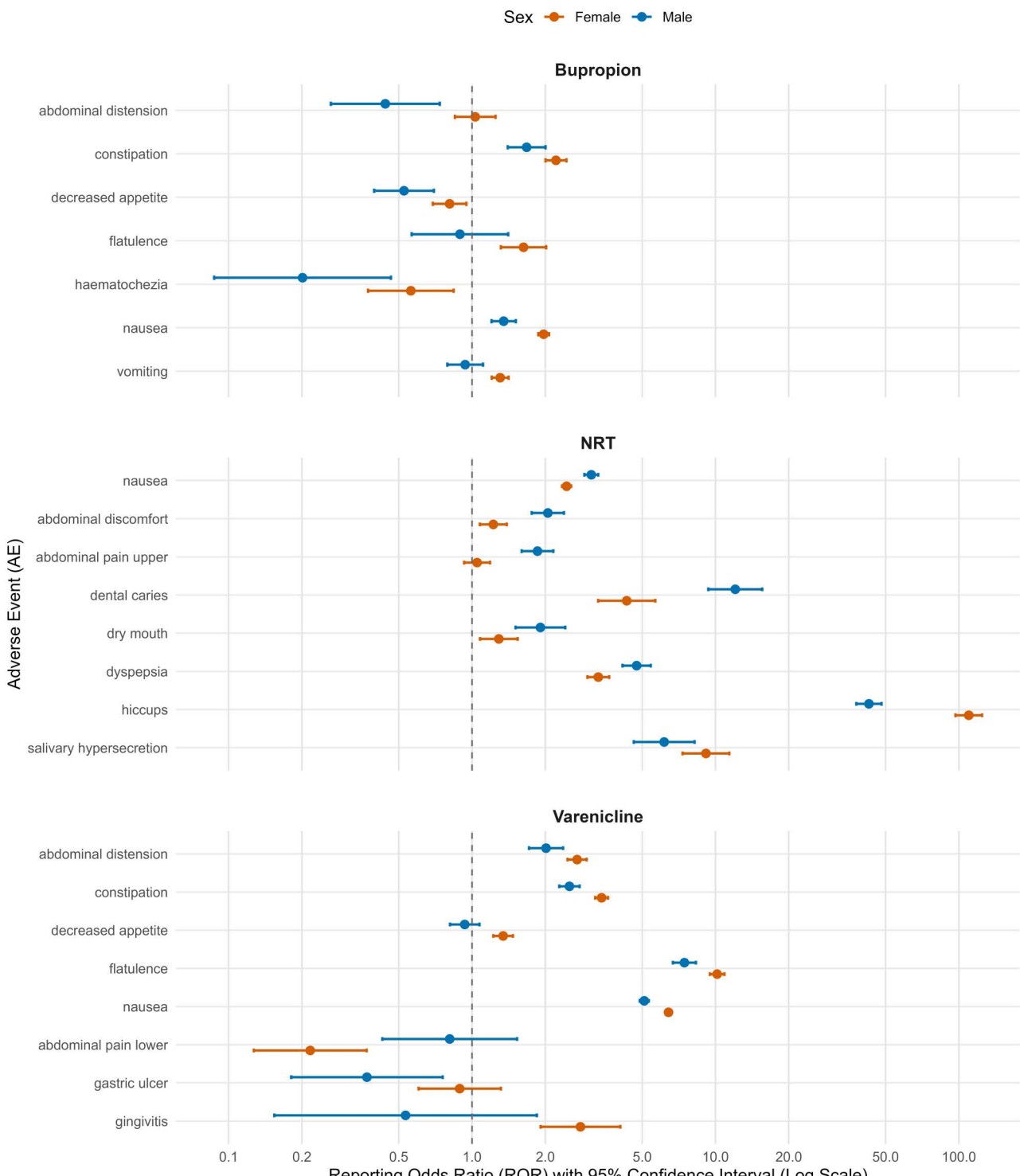

**Fig 3. Forest plots of RORs for GI AEs associated with NRT, varenicline, and bupropion.** Only events meeting the pre-specified stability filter (DE_ male + DE_female > 20) and showing significant heterogeneity by the Breslow–Day χ² test (p < 0.05) are displayed. Markers represent point estimates for male and female strata, with horizontal lines denoting 95% confidence intervals. The vertical dashed line indicates the null value (ROR = 1).

in women than in men [12,25]. Despite this, bupropion demonstrated the highest PRR signal for GI adverse events in FAERS. The effects from noradrenaline receptor activation might stimulate the GI tract in susceptible individuals, causing infrequent but severe outcomes. A strong signal was also identified for dry mouth (xerostomia), which aligns with clinical trial data where it is a well-documented side effect [26]. Unlike varenicline or NRT, bupropion provides no nicotine replacement. This means patients may undergo complete nicotine withdrawal effects, contribute to the independently oral ulcers [27].

Our analysis statistically confirmed distinct sex-specific reporting patterns for GI AEs across all three therapies. For varenicline and bupropion, a consistent trend emerged where the majority of AEs with significant heterogeneity showed stronger signals in women, including key events like nausea and constipation for both drugs (p < 0.001). In contrast, NRT exhibited a more complex profile; while signals for hiccups and salivary hypersecretion were stronger in women, a majority of the heterogeneous signals (6 of 8) were significantly stronger in men. This male-predominant pattern was evident for several GI events, including nausea and dyspepsia, and was particularly pronounced for dental caries. This general finding that women report more AEs aligns with previous research [28–30]. There are several potential explanations for these results. The average weight of female is lower than male, which lead potential greater drug exposure. Also, estrogen may impacted activity of enzyme that responsible for drug metabolism, like CYP2A6, lead to a different nicotine absorption pattern in female population [31]. Estrogen may also modulate diaphragmatic reflexes and gut motility, worsening the GI AEs [29,30].

While findings from a spontaneous reporting system cannot establish causality or guide clinical practice directly, they can highlight important areas for clinical consideration and future research. The stronger signals in women for varenicline and bupropion suggest that sex could be a key factor in treatment tolerability, warranting further investigation into sex-specific management strategies [32]. NRT users should have their technique reviewed and attend regular oral inspections to avoid gastric upset and mouth pain. Smokers with peptic-ulcer disease could benefit from starting on bupropion rather than high-dose NRT [8]. Bupropion users should be cautioned about rare but serious GI events such as dysphagia and constipation and should seek care if necessary [10,25,33]. Overall, follow-up contact within 1–2 weeks of the quit date is necessary to assess GI tolerability and reduce the failure rate of quitting smoking.

The findings of this pharmacovigilance study should be interpreted in the context of several inherent limitations of the FAERS database. As a passive surveillance system, FAERS is subject to under-reporting, and the voluntary nature of submissions can introduce reporting bias. For instance, mild or expected AEs like varenicline-related nausea may be less likely to be reported than unusual or severe events, which can impact disproportionality metrics [34,35]. Secondly, the database does not consistently record the specific route of administration for NRT, precluding a detailed analysis of how different delivery methods might influence GI AE reporting. Finally, the analysis for bupropion may be confounded by indication, as this medication is also widely prescribed for depression and other conditions; due to reporting inconsistencies, it was not feasible to systematically exclude cases unrelated to smoking cessation [10,12].

Future work should verify and extend these findings through prospective or active-surveillance studies. For example, monitoring gastrointestinal symptoms in electronic health records during quit programs could provide a more systematic assessment and help address under-reporting bias. Further research should also differentiate between types and administration routes of NRT, as delivery method may influence gastrointestinal outcomes.

In conclusion, this study characterizes the gastrointestinal safety signal profiles of NRT, varenicline, and bupropion, and explores sex-specific patterns in reporting. These findings highlight differences that warrant further investigation and may inform hypothesis-driven research on the role of sex in treatment tolerability. Ultimately, this study underscores the importance of monitoring gastrointestinal safety in smoking cessation pharmacotherapy to support successful quit attempts.

## Supporting information

**S1 Table. GI AEs lists.**
(DOCX)

**S2 Table. Proportional reporting ratios (PRRs) for all reported gastrointestinal adverse events associated with nicotine-replacement therapy (NRT), varenicline, and bupropion, presented for the overall population and stratified by sex (male and female).** Blank cells indicate no safety signal; dual filter of DE ≥ 3 and PRR ≥ 2 was used clinically significant.
(XLSX)

## Acknowledgments

We thank all colleagues and collaborators who provided valuable feedback and discussion during the preparation of this work.

## Author contributions

**Conceptualization:** Haoxiong Sun, Junchi Chen.

**Data curation:** Haoxiong Sun, Junchi Chen, Xiaoxuan Wu.

**Formal analysis:** Haoxiong Sun, Junchi Chen, Xiaoxuan Wu.

**Investigation:** Haoxiong Sun, Junchi Chen.

**Methodology:** Haoxiong Sun, Junchi Chen.

**Project administration:** Haoxiong Sun, Junchi Chen.

**Resources:** Haoxiong Sun, Junchi Chen.

**Software:** Haoxiong Sun, Junchi Chen.

**Supervision:** Haoxiong Sun.

**Validation:** Haoxiong Sun, Junchi Chen.

**Visualization:** Haoxiong Sun, Junchi Chen, Xiaoxuan Wu, Ziyan Wang.

**Writing – original draft:** Haoxiong Sun, Junchi Chen, Xiaoxuan Wu, Ziyan Wang.

**Writing – review & editing:** Haoxiong Sun, Junchi Chen, Xiaoxuan Wu, Ziyan Wang.

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
