## [Decision Letter · Decision Letter 0]

1 Sep 2025

Dear Dr. Sun,

Thank you for submitting your manuscript to PLOS ONE. After careful consideration, we feel that it has merit but does not fully meet PLOS ONE’s publication criteria as it currently stands. Therefore, we invite you to submit a revised version of the manuscript that addresses the points raised during the review process.

We look forward to receiving your revised manuscript.

Kind regards,

Jingjing Qian

Academic Editor

PLOS ONE

Journal Requirements:

2. Please amend either the title on the online submission form (via Edit Submission) or the title in the manuscript so that they are identical.

3. We notice that your supplementary tables are included in the manuscript file. Please remove them and upload them with the file type 'Supporting Information'. Please ensure that each Supporting Information file has a legend listed in the manuscript after the references list.

Reviewers' comments:

Reviewer's Responses to Questions

**Comments to the Author**

1. Is the manuscript technically sound, and do the data support the conclusions?

Reviewer #1: Yes

Reviewer #2: Partly

2. Has the statistical analysis been performed appropriately and rigorously?

Reviewer #1: Yes

Reviewer #2: No

3. Have the authors made all data underlying the findings in their manuscript fully available?

Reviewer #1: Yes

Reviewer #2: Yes

4. Is the manuscript presented in an intelligible fashion and written in standard English?

Reviewer #1: Yes

Reviewer #2: Yes

Reviewer #1: The topic of the paper is relevant, and the statistical approach is generally appropriate. However, I have two main concerns:

1.Patient counts across groups: The number of patients reported in each treatment group is inconsistent across subgroup (e.g. sex, gender). These inconsistencies should be resolved, as they raise concerns about data accuracy and can reduce confidence in the study’s findings.

2. Comparison of sex-specific signals: If you're aiming to evaluate whether the disproportionality signals (e.g., reporting odds ratios) differ by sex, consider using a formal test, such as the Breslow–Day test, rather than relying on descriptive comparisons.

Also, please double-check your formatting before submission. Several of the figures are blurry or low resolution. Clear, well-formatted visuals are important for supporting your results.

Reviewer #2: I thank the authors for their effort in mining the safety signals of smoking cessation medications using FAERS data. The authors may consider the following comments to improve the quality of the manuscript:

1. Throughout the abstract, Key points, plain language summary, and main text, the authors made a lot of statements about the prevalence of side effects for one smoking cessation medication. For example, “Varenicline produced the most gastrointestinal (GI) side- effects, especially nausea, gas, and stomach discomfort.” Please note that there is no denominator for study population or medication users in FAERS. All of the statements about adverse drug events should be made on the adverse event reports, rather than the overall drug use to avoid reporting bias, because not all adverse events are reported or submitted to FAERS. For example, GI side effects are the most commonly reported adverse events of varenicline. The authors may consider revising these statements.

2. In the Introduction, the authors mentioned “the comparative risk profiles and

phenotypic patterns among the three major smoking cessation pharmacotherapies

have not been comprehensively quantified in real-world populations”. Could the authors add 1-2 sentences to summarize the findings from published literature? RCTs and observational studies (cohort/case control studies) from large databases may be more rigorous than FAERS analysis, therefore, a strong rationale is needed to justify why FAERS data was used.

3. “The findings provide clinicians with an evidence-based appraisal of relative GI risk to support shared decision-making and patient counselling during smoking-cessation treatment.” This may be an overstatement with the limitations of FAERS data. Safety signals from FAERS do not imply and should not be interpreted as risk to guide decision making.

4. Please describe the methods of identifying reports containing smoking cessation medications (ie, search terms) from FAERS. A summary table is preferred.

5. “A signal was considered present if a given drug–event (DE) pair had at least three reports and met the threshold of PRR ≥ 2 with an accompanying chi-square (χ²) value ≥ 4.” Please add a reference for this statement.

6. The authors wanted to explore the sex-stratified safety signal, but there is no mention of methodology or statistical analysis. Please clarify

7. In the results, please add a flowchart diagram describing the steps of obtaining the final FAERS reports for studied medications from the original data files.

8. Please add counts of reports along with the % for descriptive statistics.

9. For comparing ROR/PRR between male and female, a statistical test may be needed, since the numerical difference does not imply a statistical significant.

**Do you want your identity to be public for this peer review?** For information about this choice, including consent withdrawal, please see our Privacy Policy

Reviewer #1: No

Reviewer #2: No

---

## [Author Response · Author response to Decision Letter 1]

8 Oct 2025

Response to reviewer #1

We thank Reviewer #1 for their constructive feedback and positive assessment of our work. We have addressed each of the concerns as follows:

Comment 1: “Patient counts across groups: The number of patients reported in each treatment group is inconsistent across subgroup (e.g. sex, gender). These inconsistencies should be resolved, as they raise concerns about data accuracy and can reduce confidence in the study’s findings.”

Response: We thank the reviewer for identifying this critical issue. We have meticulously re-run our data queries and aggregation scripts to resolve all inconsistencies. The patient counts presented in the main text of the Results section and in Table 2 are now fully harmonized and accurate. The total number of reports and the sums of all subgroups now align perfectly.

(See Results section and Table 2)

Comment 2: “Comparison of sex-specific signals: If you're aiming to evaluate whether the disproportionality signals (e.g., reporting odds ratios) differ by sex, consider using a formal test, such as the Breslow–Day test, rather than relying on descriptive comparisons.”

Response: We agree completely with the reviewer that a formal statistical test is essential for this comparison. This was a significant oversight in our original draft. We have now incorporated the Breslow–Day test to formally assess the homogeneity of odds ratios between male and female strata.

The methodology for this test has been added to a new subsection in the Methods section titled "Sex-stratified disproportionality and Breslow–Day test". The results of this analysis, including p-values, are now presented in Table 4 and visualized in the new forest plots in Figure 3. This addition has substantially improved the rigor of our study.

(See Methods section, and Results section, Table 4 and Figure 3)

Comment 3: “Also, please double-check your formatting before submission. Several of the figures are blurry or low resolution. Clear, well-formatted visuals are important for supporting your results.”

Response: We sincerely apologize for the poor quality of the figures in the original submission. We have recreated all figures at a high resolution (300 DPI) to ensure that all figure files now meet the journal's technical requirements for clarity, size, and format.

Response to reviewer #2

We are grateful to Reviewer #2 for their thorough and detailed feedback, which has led to substantial improvements throughout the manuscript. We have addressed all points raised below.

Comment 1: “Throughout the abstract, Key points, plain language summary, and main text, the authors made a lot of statements about the prevalence of side effects... Please note that there is no denominator... All of the statements... should be made on the adverse event reports, rather than the overall drug use to avoid reporting bias...”

Response: This is an essential point regarding the limitations of FAERS data. We have carefully revised the entire manuscript (including the Abstract, Key Points, and Plain Language Summary) to ensure our language is precise.

We now consistently use terms such as “reporting profiles,” “disproportionality signals,” and “proportion of reports” instead of terms that might imply prevalence or causal risk (e.g., “prevalence,” “risk,” or “produced”).

(See revisions throughout the Abstract, Results, and Discussion sections)

Comment 2: “In the Introduction, the authors mentioned ‘the comparative risk profiles... have not been comprehensively quantified...’ Could the authors add 1-2 sentences to summarize the findings from published literature? ...a strong rationale is needed to justify why FAERS data was used.”

Response: We thank the reviewer for this suggestion. We have revised the Introduction to strengthen our rationale. We now explicitly state the limitations of controlled trials and other observational studies (e.g., their inability to detect rare signals or reflect real-world heterogeneity) to better justify the unique value of a large-scale pharmacovigilance study using the FAERS database for hypothesis generation.

(See Introduction)

Comment 3: “‘The findings provide clinicians with an evidence-based appraisal of relative GI risk...’ This may be an overstatement with the limitations of FAERS data. Safety signals from FAERS do not imply and should not be interpreted as risk to guide decision making.”

Response: We agree that our original statement was an overstatement. We have toned down the language in the Discussion and Conclusion to reflect the hypothesis-generating nature of our findings. We now state that these results cannot establish causality or directly guide clinical practice but can highlight areas for future research and inform clinical consideration.

(See Discussion and Conclusion)

Comment 4: “Please describe the methods of identifying reports containing smoking cessation medications (ie, search terms) from FAERS. A summary table is preferred.”

Response: As requested, we have added Table 1, which provides a comprehensive list of the search terms (both generic and brand names) used to identify reports for NRT, varenicline, and bupropion in the FAERS database.

(See Methods section and Table 1)

Comment 5: “‘A signal was considered present if a given drug–event (DE) pair had at least three reports and met the threshold of PRR ≥ 2 with an accompanying chi-square (χ²) value ≥ 4.’ Please add a reference for this statement.”

Response: We have now added a citation for this established signal detection criterion in pharmacovigilance.

(See Methods section, "Data Mining" subsection)

Comment 6 & 9: “The authors wanted to explore the sex-stratified safety signal, but there is no mention of methodology or statistical analysis. Please clarify” and “For comparing ROR/PRR between male and female, a statistical test may be needed...”

Response: We apologize for this critical omission. As also noted by Reviewer #1, we have now added a detailed methodology for the sex-stratified analysis. A new subsection, "Sex-stratified disproportionality and Breslow–Day test," has been added to the Methods section. This section describes how we performed the stratification and used the Breslow–Day test to formally assess for statistically significant differences in reporting between sexes. The results are presented in Table 4 and Figure 3.

(See Methods section, and Results section, Table 4 and Figure 3)

Comment 7: “In the results, please add a flowchart diagram describing the steps of obtaining the final FAERS reports...”

Response: We thank the reviewer for this suggestion to improve clarity. We have added a flowchart (Figure 1) that visually details the entire process of case selection, from the initial number of reports in the database to the final dataset used in our analysis.

(See Results section, Figure 1)

Comment 8: “Please add counts of reports along with the % for descriptive statistics.”

Response: We have revised our descriptive statistics to include both absolute counts (n) and percentages (%). This format has been applied to the main text of the Results section and is clearly presented in Table 2, where the header clarifies that values are "n (% of all reports for that drug)."

(See Results section and Table 2)

---

## [Decision Letter · Decision Letter 1]

20 Oct 2025

Sex-stratified pharmacovigilance of gastrointestinal events associated with first-line smoking-cessation medicines: insights from the FAERS database

PONE-D-25-33119R1

Dear Dr. Sun,

We’re pleased to inform you that your manuscript has been judged scientifically suitable for publication and will be formally accepted for publication once it meets all outstanding technical requirements.

Kind regards,

Jingjing Qian

Academic Editor

PLOS ONE

Additional Editor Comments (optional):

Thanks for addressing all comments raised by both reviewers. The text under "Acknowledgments" seems not relevant or out of place (line 378). Please review and fix it during the proofing process.

Reviewers' comments:

Reviewer's Responses to Questions

**Comments to the Author**

Reviewer #1: All comments have been addressed

2. Is the manuscript technically sound, and do the data support the conclusions?

Reviewer #1: Yes

3. Has the statistical analysis been performed appropriately and rigorously?

Reviewer #1: Yes

4. Have the authors made all data underlying the findings in their manuscript fully available?

Reviewer #1: Yes

5. Is the manuscript presented in an intelligible fashion and written in standard English?

Reviewer #1: Yes

Reviewer #1: (No Response)

**Do you want your identity to be public for this peer review?** For information about this choice, including consent withdrawal, please see our Privacy Policy

Reviewer #1: No

---

## [Editor Report · Acceptance letter]

PONE-D-25-33119R1

PLOS ONE

Dear Dr. Sun,

I'm pleased to inform you that your manuscript has been deemed suitable for publication in PLOS ONE. Congratulations! Your manuscript is now being handed over to our production team.

Kind regards,

on behalf of

Dr. Jingjing Qian

Academic Editor

PLOS ONE